# Using Waste Sulfur from Biogas Production in Combination with Nitrogen Fertilization of Maize (*Zea mays* L.) by Foliar Application

**DOI:** 10.3390/plants10102188

**Published:** 2021-10-15

**Authors:** Petr Škarpa, Jiří Antošovský, Pavel Ryant, Tereza Hammerschmiedt, Antonín Kintl, Martin Brtnický

**Affiliations:** 1Department of Agrochemistry, Soil Science, Microbiology and Plant Nutrition, Mendel University in Brno, Zemědělská 1, 613 00 Brno, Czech Republic; petr.skarpa@mendelu.cz (P.Š.); pavel.ryant@mendelu.cz (P.R.); tereza.hammerschmiedt@mendelu.cz (T.H.); antonin.kintl@mendelu.cz (A.K.); martin.brtnicky@mendelu.cz (M.B.); 2Agricultural Research, Ltd., Zahradní 400/1, 664 41 Troubsko, Czech Republic

**Keywords:** chlorophyll content, fluorescence parameters, plant weight, plant nutrient content, nitrogen use efficiency

## Abstract

In Europe, mainly due to industrial desulfurization, the supply of soil sulfur (S), an essential nutrient for crops, has been declining. One of the currently promoted sources of renewable energy is biogas production, which produces S as a waste product. In order to confirm the effect of the foliar application of waste elemental S in combination with liquid urea ammonium nitrate (UAN) fertilizer, a vegetation experiment was conducted with maize as the main crop grown for biogas production. The following treatments were included in the experiment: 1. Control (no fertilization), 2. UAN, 3. UANS1 (N:S ratio, 2:1), 4. UANS2 (1:1), 5. UANS3 (1:2). The application of UAN increased the N content in the plant and significantly affected the chlorophyll content (N-tester value). Despite the lower increase in nitrogen (N) content and uptake by the plant due to the application of UANS, these combinations had a significant effect on the quantum yield of PSII. The application of UANS significantly increased the S content of the plant. The increase in the weight of plants found on the treatment fertilized with UANS can be explained by the synergistic relationship between N and S, which contributed to the increase in crop nitrogen use efficiency. This study suggests that the foliar application of waste elemental S in combination with UAN at a 1:1 ratio could be an effective way to optimize the nutritional status of maize while reducing mineral fertilizer consumption.

## 1. Introduction

One of the principles of the European Green Deal is the proposal of greenhouse gas emissions cut by at least 55% by the year 2030, which should set Europe to a path to becoming climate-neutral by the year 2050 [1]. According to the European Biogas Association (EBA), biogas, biomethane, and other renewable gases will play a key role in helping Europe’s transition to a clean energy system [2], and the European Commission’s strategies promise targeted support for biogas in the revised Renewable Energy Directive and gas legislation. EBA, Eurogas, and the Gas for Climate consortium are calling for an EU-wide renewable target of at least 11%. The annual production of biogas in Europe reaches 15.8 bcm and is relatively stable with a total of 18,943 biogas plants according to the EBA [3].

A biogas plant produces biogas, which can then be used for the cogeneration of electricity and heat. Biogas is a mixture of methane, carbon dioxide, and other components such as hydrogen sulfide (H_2_S) [4]. The biogas must be pretreated before use. The first step of the purification process is the removal of H_2_S, which is corrosive and harmful to health [5,6]. Biogas production is thus associated with the production of waste products. The utilization of waste sulfur obtained from the purification process seems to be promising from the point of view of plant nutrition and especially from the economic aspect of biogas production [7,8] and sulfur deficiency in the environment.

European SO_2_ emissions have been reduced by 70–80% since 1990 [9,10]. According to results of Engardt et al. [11], sulfur deposition in Europe will decrease until, at least, 2050. For example, in the Czech Republic, atmospheric sulfur deposition is about 5 kg/ha per year [12], so there is a shortage of sulfur in the soil, as it has been presented by many authors [13,14,15,16,17,18]. According to Zbíral et al. [19], a statistically highly significant decrease in the soil S content caused by reduction of SO_2_ emissions in the long-term field experiments in Czech Republic from 33 mg/kg in 1981 to 8 mg/kg in 2017. Therefore, it is necessary to pay special attention to fertilization by sulfur in addition to the other essential nutrients, especially because of the increased cultivation of crops with high sulfur requirements [19,20]. Sulfur in plants is essential for the synthesis of cysteine, methionine, and some vitamins [21]. The deficiency of sulfur in maize as the main crop for biogas production not only reduces yield but also quality parameters such as the content of starch, carbohydrates, and proteins [22]. Sulfur is usually applied in the form of mineral fertilizers, and co-application with nitrogen is recommended by many authors as these nutrients have been proven to have good synergy [23,24]. Salvagiotti and Miralles [25] showed that S addition increased the biomass and grain yield of cereal and the positive interaction of N and S, which resulted in a greater nitrogen use efficiency. A shortage of S supply also lowers the utilization of nitrogen and results in a deterioration in crop quality [26]. As sulfur is an essential constituent of enzymes involved in nitrogen metabolism, its deficiency could lead to a decrease in N assimilation [27,28]. Some reports have shown the accumulation of nitrates in S-deficient plants [29]. In addition, Haneklaus et al. [30] reported that each kg of S deficit causes 15 kg of nitrogen to be lost in the environment. Maize is an important crop that, despite its relatively low sulfur requirements, is severely affected by its deficiency [31,32].

Nitrogen is essential for plants in terms of biomass and yield production [33]. In addition to the conventional nitrogen fertilization of the soil, the nutritional status of the plants can be optimized by foliar fertilization during the plant growth [34]. Foliar fertilization could be used under farming conditions as a quick correction for unexpected nutrient deficiencies, for the late supply of N (and another nutrients) during advanced growth stages, and as a preventive measure against unsuspected (or hidden) deficiencies [35,36,37,38]. The foliar application of nutrients is also recommended when the soil or the plant conditions limit the availability of some nutrients [39] and is appropriate under conditions when high loss rates of soil-applied nutrients may occur [40]. For example, the foliar application of nitrogen has significantly improved the grain yield of maize [41] and other cereal crops [42].

The aim of this study was to verify the effect of the foliar application of waste elemental sulfur from biogas production in combination with conventional liquid fertilizers UAN applied in different ratios. Such a reutilization of waste sulfur from biogas plants back in agriculture is suitable from the economic aspect of biogas purification and waste management. The application of this sulfur could help to reduce the consumption of mineral fertilizers and, at the same time, address the deficient sulfur content in the soil and plants.

## 2. Results and Discussion

The application of UAN fertilizer alone and in combination with sulfur increased the chlorophyll content (N-tester value) in maize leaves compared to the unfertilized control. The increase in chlorophyll was evident at both monitoring terms (t1 and t2), while the differences between the control (N-unfertilized treatment) and the N (UAN) and NS (UANS1-3)-fertilized treatments increased over time (Table 1).

The N-tester values were significantly correlated with the rate of nitrogen applied in fertilizers at both terms, as presented in Figure 1. The results agree with several studies that have reported a strong correlation between chlorophyll content and the amount of nitrogen in leaves [43,44,45,46,47].

Nitrogen is part of the enzymes associated with chlorophyll synthesis [48] and the chlorophyll concentration reflects relative crop N status. Statistically significant highest N-tester values were found for the treatment fertilized with UAN applied without sulfur (UAN) on both measurement terms (t1; t2). The highest nitrogen dose was applied on this treatment. The application of UAN in combination with elemental sulfur (UANS1–3) significantly increased the N-tester value compared to the unfertilized (control) treatment, but the level of chlorophyll content did not reach the values found in plants fertilized with UAN alone. While, in the first measurement term (t1), the highest N-tester value was found for the UANS2 treatment (Table 1), in term t2, the N-tester values were in direct dependence on the nitrogen doses contained in the UAN–sulfur mixture. The N-tester values found at both terms (t1; t2) were significantly correlated with the plant nitrogen content detected at term t3 (r = 0.711, *p* ˂ 0.001; r = 0.707, *p* ˂ 0.001, respectively). Evaluation of the nutritional status after the joint application of nitrogen and sulfur using the N-tester was also performed on several dates by Lacroux et al. [49], and their results showed a significant increase in measured values compared to the control, with the highest values achieved by the joint foliar application of N and S.

The ability of the photosystem II to absorb radiation is expressed by the variable chlorophyll fluorescence for dark-adapted leaves (*F_v_*). The more radiation a plant can absorb, the more radiation the plant can use for photosynthesis. Although the ability of the plant tissue to absorb radiation decreased over time (comparison of *F_v_* levels between t1 and t2), this decrease was not significant for the UAN and UAN combination with sulfur. A significant reduction in *F_v_* values was only observed in the unfertilized treatment (Figure 2). Even though the treatment with the highest sulfur dose (UANS3) showed the lowest *F_v_* values, the results showed that the decrease in *F_v_* between terms t1 and t2 was smallest on this treatment. Nitrogen deficiency decreases the photosynthetic assimilation capacity of CO_2_ of plant leaves, leading to decreases in light-saturated photosynthetic rates [50]. In addition, Ciompi et al. [51] and Jin et al. [52] reported a positive correlation between the nitrogen content in the plant tissue of leaves and photosynthetic capacity.

After dark adaptation of the maize leaves, the maximum photosynthetic capacity (*Φ_PSII_*) was estimated as the quotient between variable and maximum fluorescence (*F_v_*/*F_m_*). The quantum yield, which indicates the actual capacity for photochemical processes by the availability of reaction centers of the photosystem II (PSII), was significantly (*p* ≤ 0.05) influenced by the fertilizer application (Figure 3). It is clear that nitrogen significantly affects photosynthesis and chlorophyll fluorescence of the plant. This was demonstrated by the response of maize to nitrogen fertilization in a study by Ahmad et al. [53], in which the effect of nitrogen application increased the electron transport rate, photochemical quenching coefficient, variable fluorescence, maximal quantum yield, and effective quantum yield of PSII photochemistry. A significant increase in *Φ_PSII_* values in three maize varieties due to a high nitrogen dose was demonstrated by Jin et al. [52]. Reductions in the quantum yield of PSII electron transfer due to nitrogen deficiency were also described by Nunes et al. [54] and Verhoeven et al. [55]. In our study, the values of *Φ_PSII_* were decreased over time regardless of fertilization treatment. The highest value of *Φ_PSII_* was determined after the application of UAN with the highest elemental sulfur content (UANS3). These results contradict the above studies, but, on the other hand, they show a positive effect of applied sulfur on nitrogen utilization and its use by the plant. A high linear dependence between the efficiency of carbon fixation and quantum yield value was presented by Fryer et al. [56].

The rate of fluorescence decline (*R_Fd_*), an empirical parameter for the quantification of plant vitality under tested conditions, was measured. In contrast to the values of the variable chlorophyll fluorescence (*F_v_*) and quantum yield of PSII (*Φ_PSII_*), the rate of fluorescence decline was not statistically significantly affected by foliar fertilization. Only at term t2 did the *R_Fd_* value of plants grown on the UANS2 treatment decrease significantly below the control level, but no trend in the decrease in *R_Fd_* due to UAN fertilization in combination with elemental sulfur was observed (Figure 4).

The average dry weight of the above-ground biomass (AGB) of plants determined on the 35th day after the foliar application of fertilizer (t3) is shown in Figure 5. The highest plant dry weight was found for the treatment fertilized with UAN, which provided the most nitrogen to the plants. The dry weight of plants produced on this treatment was 2.4 times higher compared to the unfertilized Control. The dry weight of plants fertilized with the UANS fertilizer combination ranged from 17.44 to 17.84 g/plant and was not statistically different from the UAN treatment (Figure 5). A significant effect of foliar nitrogen application on plant dry matter yield has been demonstrated in the available literature [57,58,59], in agreement with our results. The increase in plant weight due to foliar sulfur fertilization was also documented. Perveen et al. [60] observed a significant increase in root and shoot biomass and root and shoot length of maize grown under salinity conditions due to the foliar application of different sulfur compounds. An increased barley yield after elemental sulfur application was described by Grzebisz and Przygocka-Cyna [61] in their long-term experiment. A positive effect of the foliar application of sulfur on canola pods formation and subsequent seed yield was demonstrated by Khalid et al. [62].

The UAN fertilizer application significantly increased the nitrogen content of maize leaves. The highest N content, 10.3 g/kg DM, was found in leaves after the application of UAN fertilizer alone (Table 2). There was no significant difference in plant N content among treatments fertilized with a mixture of UAN and elemental sulfur (UANS1-3), but the data showed a relative increase in N content with sulfur rate. An increased leaf N concentration following sulfur fertilization has also been described [31,63]. An increase in the nitrogen content of wheat grain, due to the foliar application of sulfur, was observed by Tea et al. [64] and Rossini et al. [65]. This effect could be due to a better assimilation of foliar-applied N and S compared to their soil-applied counterparts.

Sutar et al. [22] described the critical sulfur concentration in dry matter of maize leaves as 1.5 g/kg DM. The sulfur content in the ABG of maize plants ranged from 2.8 to 3.6 g/kg DM (Table 2). Its content in the ABG of plants grown on the treatments fertilized with a mixture of UAN and elemental sulfur (UANS1–3) was identical to that of unfertilized plants (Control). Only in the nitrogen-fertilized treatment (UAN) was the amount of sulfur significantly lowest (Table 2). This fact is not only related to the absence of sulfur in the fertilizer, but it can also be explained by the dilution of nutrients in the maize plant tissue that occurred as a result of the increase in DM weight of AGB on this treatment (Figure 5). Therefore, the nutrient uptake by the plant was calculated as a more appropriate parameter expressing the nutritional status of the plants (Figure 6). Nutrient uptake is the relationship between the DM weight of AGB and its nutrient content, expressed in g of nutrient per plant (g/plant). Logically, the highest nitrogen uptake was recorded in the UAN-fertilized treatment, i.e., the treatment with the highest applied nitrogen rate. Even though nitrogen uptake by plants was not significantly different among the treatments fertilized with UAN and elemental sulfur mixtures, plants fertilized with fertilizers containing a higher proportion of elemental sulfur (UANS2 and UANS3) showed a higher uptake of nitrogen by plant AGB. A positive significant interaction between nitrogen and sulfur uptake and utilization was confirmed.

The N:S ratio of the plant may also be an interesting indicator of nutritional status, as reported by some authors [31,66]. The principle behind this assessment is the fact that plants need a balanced amount of nitrogen and sulfur for proper amino acid synthesis. Therefore, nitrogen-to-sulfur ratios above a N:S ratio threshold indicate S deficiency [67]. A possible disadvantage of this assessment is the decreasing value of the N:S ratio during the growing season, as reported, for example, by Calvo et al. [68,69] or Scherer [70]. A 15-19:1 N:S ratio has been reported as a limiting ratio for cereals at the time of tillering [71], and an ideal N:S ratio for the optimum growth and development of maize is 15:1 [72]. The observed N:S ratio (Table 2) indicated that the sulfur contained in maize was not deficient in any of the fertilization treatments. From the ratios obtained, it is possible to observe the already described trend, where the highest ratio of nitrogen and sulfur was logically found on the treatment fertilized only with UAN fertilizer. In contrast to our study, significant changes in the N:S ratio after sulfur application were observed [73,74]. However, they agreed that an increase in the sulfur content of the plant does not necessarily predict increased yield. Sutradhar et al. [31] also confirmed the same conclusion.

The previously mentioned synergism between nitrogen and sulfur can be documented by crop nitrogen use efficiency. The nitrogen supplied by foliar nutrition from fertilizer applied without sulfur addition (UAN) was utilized by the plant at 30.5% (Table 3). A similar level of NUE_Crop_ was found on the treatment fertilized with the lowest sulfur fertilizer mixture (UANS1), whereas an increase in the proportion of sulfur in the fertilizer mixture increased nitrogen use efficiency. The relationship between nitrogen recovery from applied fertilizers and the dose of sulfur applied by the fertilizer mixture was statistically significant (NUE_Crop_ = 22.9 + 0.171 × sulfur dose, r = 0.709; *p* = 0.002).

In agreement with our results, several studies showed that sulfur fertilization may increase NUE [75,76,77]. As sulfur is an essential constituent of enzymes involved in nitrogen metabolism [78], its deficiency may lead to ineffective utilization of the nitrogen content in plant [79,80]. An increase in nitrogen uptake by maize plants due to graded doses of foliar sulfur application was presented by Sarfaraz et al. [81].

## 3. Materials and Methods

### 3.1. Experimental Methodology, Plant Material, and Growth Conditions

The pot vegetation experiment was established in the vegetation hall of the Biotechnological house at Mendel University in Brno located at 49°21′03′′ N and 16°61′38′′ E. Mitscherlich pots (STOMA GmbH, Siegburg, Germany) were filled with 6.5 kg of air-dried and sieved soil (2 cm diameter sieve). Properties of the soil used in the pot experiment are shown in Table 4.

The maize (*Zea mays* L.), cultivar SY ORPHEUS (Syngenta Czech s.r.o., Prague, Czech Republic), was chosen for this study. Four seeds of maize were sown to a 4 cm depth in each pot. The number of plants in each pot was reduced to two plants per pot two weeks after the sowing.

The pot experiment was carried out under seminatural conditions in the outdoor vegetation hall under a rain shelter. The air temperature, air humidity, and solar radiation during the maize growing season are shown in Figure 7. After a cooler April (11.8 °C) and May (12.6 °C), a warming period occurred at the beginning of June (22.6 °C), which lasted until the end of the experiment (average air temperature in July was 19.6 °C). The relative air humidity fluctuated evenly between 40 and 90% during the experiment. Global solar radiation also fluctuated over time depending on weather conditions, with levels increasing slightly during the experiment (April: 16.9; July: 22.1 MJ/m^2^). A controlled watering regime identical for all treatments (pots) was used in the experiment. Plants were watered to 70% of the maximum water holding capacity throughout the growing season. The pots were watered by hand with demineralized water on the soil surface.

Liquid urea ammonium nitrate fertilizer (UAN; 30% total N–15% N-NH_2_, 7.5% N-NO_3_, 7.5% N-NH_4_) was applied to maize plants in combination with waste elemental sulfur suspension (12% S^0^ suspension) in the ratios shown in Table 5. The sulfur suspension was obtained by the desulfurization of biogas using the Thiopaq^®^ scrubber (Paques, Balk, The Netherlands), which works by washing the raw biogas with a slightly alkaline solution (pH 8–9) and the subsequent biological oxidation of sulfides to elemental sulfur. The elemental sulfur particle size in the suspension was less than 60 µm (96.9% of the particles). Each of the treatments was established in eight replicates (pots). The pots were placed randomly in the vegetation hall under the rain shelter.

The foliar application was carried out on the plant development stage of the 4th true leaf unfolded. The application of 3 mL of fertilizer mixture per pot of each treatment was used. Fertilizers were evenly applied using a pressurized hand pump sprayer (DPZ 1500, ProGlass, Weilheim an der Teck, Germany). The mixture of the waste elemental sulfur with the UAN fertilizer was mixed prior to application to ensure that the elemental sulfur was evenly distributed in the fertilizer mixture and applied uniformly.

During the maize vegetation, chlorophyll content (N-tester value) and chlorophyll fluorescence parameters were evaluated. The measurements were performed 7 (t1) and 21 days (t2) after the foliar application. The weight of dry matter (DM) of maize plant AGB, the content and ratio of nutrients (N and S) in maize plant AGB, and their uptake by plants were determined 35 days after the foliar application of fertilizer mixtures (t3). The schedule of the experiment is shown in Table 6.

### 3.2. Determination of Plant Growth and Development Parameters

#### 3.2.1. Chlorophyll Content in Plant Leaf (N-Tester Value)

The chlorophyll content of maize leaves was measured using a Yara N-tester (Yara International ASA, Oslo, Norway). The chlorophyll content was expressed as “N-tester value.” Measurement was performed at a wavelength range of 650–940 nm [85]. Eight plants were assessed in each treatment in both terms. The measurement of chlorophyll content was performed on the 5th (t1) and 6th true leaves (t2), and the value of the chlorophyll content of each plant was the mean of 60 measurements.

#### 3.2.2. Chlorophyll Fluorescence Parameters

To determine the photochemical efficiency of photosystem II, selected fluorescence parameters of chlorophyll were measured in maize plant. The tested parameters were measured with the PAR-FluorPen FP 110-LM/S (Photon Systems Instruments, Drásov, Czech Republic) and evaluated using the FluorPen 1.1 software [86]. Measurements of fluorescence parameters were carried out on identical leaves (5th and 6th true leaves) at identical terms (t1 and t2) as for chlorophyll content determination. Maize leaves were dark-adapted for 25 min before measurement. The protocol for measuring the fluorescence parameters of chlorophyll is shown in Table 7.

The variable fluorescence of the dark-adapted leaves (*F_v_*), quantum yield of photosystem II (*Φ_PSII_*), and chlorophyll fluorescence decrease ratio (*R_Fd_*) were determined (Table 8).

#### 3.2.3. Determination of Weight Biomass, Nutrient Contents and Uptake, and Nitrogen Use Efficiency

The ABG of maize plants was harvested on 22 July 2019 (t3). The ABG was then oven-dried at 60 °C for the first two hours. The temperature was then reduced to 45 °C where the samples were kept for 72 h. The dry weight of ABG was determined using a laboratory-scale PCB Kern (KERN & Sohn GmbH, Balingen, Germany). Then, the dried ABG was crushed and homogenized by the grinder Grindomix GM200 (Retsch GmbH, Haan, Germany). The HNO_3_/H_2_O_2_ [90] digestion of biomass was achieved using a microwave digestion system in ETHOS 1 (Milestone Srl, Sorisole, Italy). Subsequently, the nutrient content (Table 9) and nutrient ratio were determined, and crop nitrogen use efficiency (NUE_Crop_) was calculated by the relationship NUE_Crop_ = N yield/N input · 100 (N yield = nitrogen uptake by plants (mg/pot); N input = nitrogen applied by foliar fertilizers (mg/pot)) [91]. In the calculation of NUE_Crop_, the nitrogen uptake by plants grown on the fertilized treatments (N yield) was subtracted from the nitrogen uptake observed on the control treatment (this amount of nitrogen characterized the natural soil supply).

### 3.3. Statistical Data Analysis

The effect of the foliar application of fertilizer mixtures on the plant growth and development parameters was statistically analyzed by the STATISTICA 12 program (TIBCO Software, San Jose, CA, USA) [94]. The effect of the foliar application on the N-tester value, chlorophyll fluorescence parameters, dry weight of ABG, and nutrient content ABG of maize was analyzed separately for each treatment of the experiment. The normality was checked using the Shapiro–Wilk test, and the homogeneity was verified by the Levene test at *p* ≤ 0.05. The effect of fertilization was analyzed using ANOVA. Fisher’s LSD test (*p* ≤ 0.05) was used to determine any statistically significant differences between the means of treatments.

## 4. Conclusions

Nitrogen plays an important role in maize nutrition, contributes to chlorophyll formation, and significantly influences the photosynthetic activity of plants. The result of the vegetation experiment showed that the efficiency of mineral nitrogen fertilization can be increased by the foliar application of liquid fertilizers with sulfur addition. The optimal adjustment of the ratio of applied nutrients (N and S) improves the nutritional status of the plants and allows the reduction in their doses while minimizing the environmental risks associated with fertilization. The foliar application of UAN fertilizer in combination with elemental sulfur from biogas production in a 1:1 ratio seems to be a sensible way to optimize the nutritional status of maize, both for the economics of biogas purification, when the waste sulfur is reused as a fertilizer, and for environmental reasons. However, verification of the results obtained from the pot experiment in field trials will be necessary.

## Figures and Tables

**Figure 1 plants-10-02188-f001:**
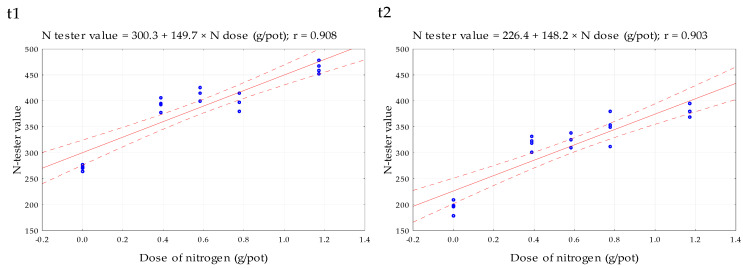
Dependence of N-tester value on nitrogen dose. The measurements were carried out on the 1st (t1) and 2nd (t2) growth stages of maize.

**Figure 2 plants-10-02188-f002:**
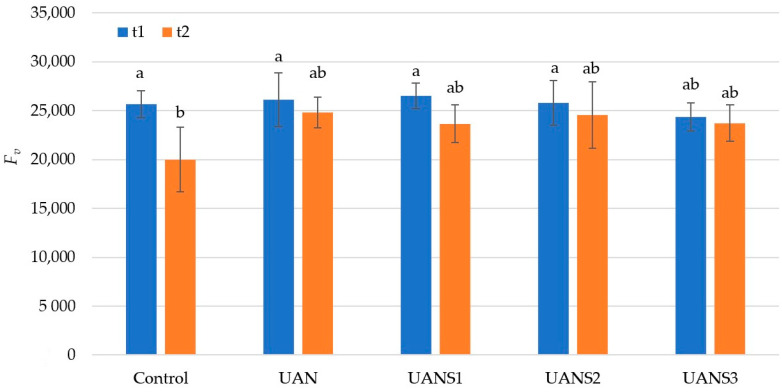
Variable fluorescence (*F_v_*) value after the foliar application of fertilizers. The measurements were carried out on two growth stages of maize (t1 and t2). The values represent the arithmetic mean (*n* = 8); the bars represent the standard deviation of the mean. There are no statistical differences between columns with the same letters (Fisher’s LSD test, *p* ˂ 0.05).

**Figure 3 plants-10-02188-f003:**
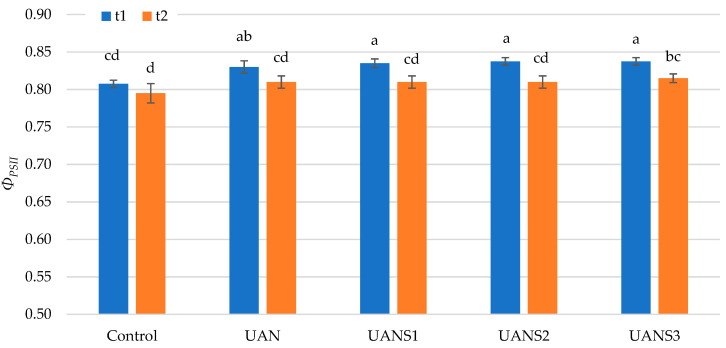
The effect of the foliar application of fertilizers on the quantum yield of PSII photochemistry (*Φ_PSII_*). The measurements were carried out on two growth stages of maize (t1 and t2). The values represent the arithmetic mean (*n* = 8); the bars represent the standard deviation of the mean. There are no statistical differences between columns with the same letters (Fisher’s LSD test, *p* ˂ 0.05).

**Figure 4 plants-10-02188-f004:**
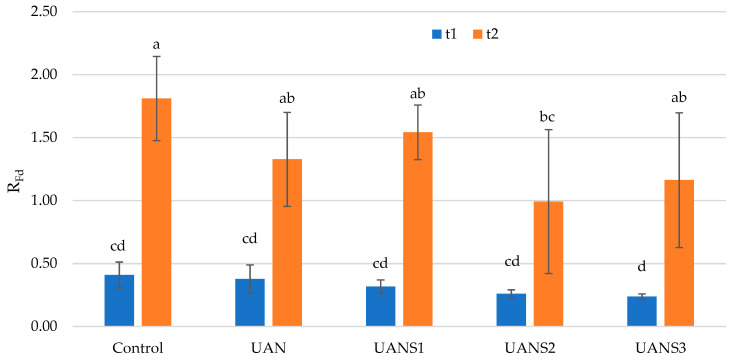
Fluorescence decrease ratio (*R_Fd_*) in maize leaves after the foliar application of fertilizer. The measurements were carried out on two growth stages of maize (t1 and t2). The values represent the arithmetic mean (*n* = 8); the bars represent the standard deviation of the mean. There are no statistical differences between columns with the same letters (Fisher’s LSD test, *p* ˂ 0.05).

**Figure 5 plants-10-02188-f005:**
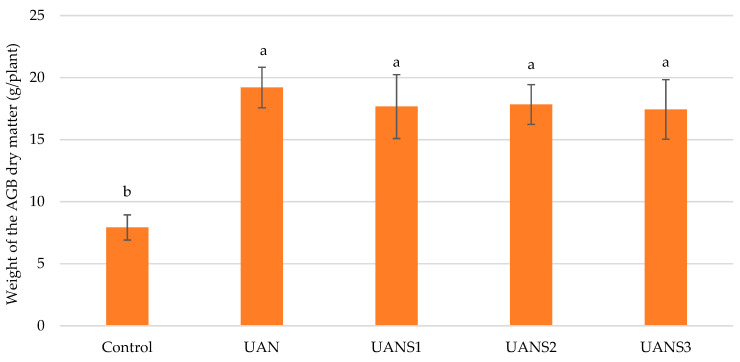
Weight of dry matter above-ground biomass of maize after the foliar application of fertilizer. The measurements were taken at the end of experiment (t3). The values represent the arithmetic mean (*n* = 8); the bars represent the standard deviation of the mean. There are no statistical differences between columns with the same letters (Fisher’s LSD test, *p* ˂ 0.05). AGB—above-ground biomass.

**Figure 6 plants-10-02188-f006:**
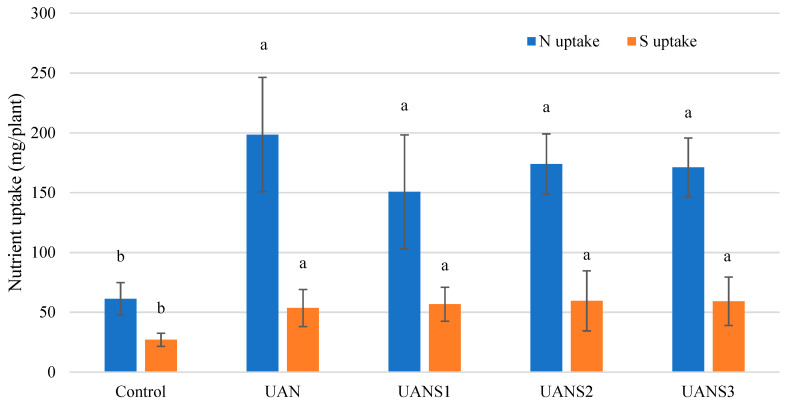
Nitrogen and sulfur uptake by above-ground plant biomass (mg/plant). The measurements were taken at the end of the experiment (t3). The values represent the arithmetic mean (*n* = 8); the bars represent the standard deviation of the mean. There are no statistical differences between columns with the same letters (Fisher’s LSD test, *p* ˂ 0.05).

**Figure 7 plants-10-02188-f007:**
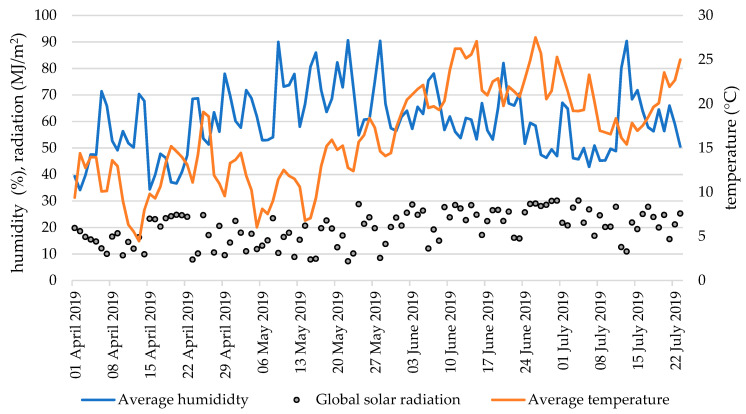
The average daily temperature (°C), relative humidity (%), and global solar radiation (MJ/m^2^) in the vegetation hall during the experiment.

**Table 1 plants-10-02188-t001:** The effect of the foliar fertilizer application on chlorophyll contents (N-tester value).

Treatment	t1		t2	
	N-Tester Value	Rel. %	N-Tester Value	Rel. %
Control	271 ± 5 ^f^	100.0	196 ± 12 ^g^	100.0
UAN	465 ± 11 ^a^	171.6	379 ± 13 ^cd^	193.4
UANS1	397 ± 14 ^bc^	146.5	349 ± 28 ^de^	178.0
UANS2	414 ± 11 ^b^	152.8	325 ± 12 ^e^	165.8
UANS3	393 ± 12 ^bc^	145.0	319 ± 13 ^e^	162.8

The values in the table represent the arithmetic mean (*n* = 8) ± SD (standard deviation). The same letters next to the numbers describe no statistically significant differences between the treatments (Fisher’s LSD test, *p* ˂ 0.05). The relative expression of the values is shown in the column marked Rel. % (Control = 100%). The measurements were performed at two growth stages, t1 (5th true leaf) and t2 (6th true leaf).

**Table 2 plants-10-02188-t002:** Nutrient content and nutrient uptake by DM of AGB and the N:S ratio.

Treatment	Nitrogen		Sulfur		N:S Ratio
	g/kg DM	Rel. %	g/kg DM	Rel. %	
Control	7.7 ± 0.8 ^b^	100.0	3.4 ± 0.6 ^a^	100.0	2.3 ± 0.5 ^b^
UAN	10.3 ± 1.8 ^a^	134.0	2.8 ± 0.6 ^b^	80.9	3.8 ± 0.4 ^a^
UANS1	9.3 ± 0.6 ^a^	121.6	3.6 ± 0.4 ^a^	104.4	3.1 ± 0.8 ^ab^
UANS2	9.7 ± 0.7 ^a^	127.1	3.3 ± 1.3 ^a^	97.1	3.4 ± 1.6 ^ab^
UANS3	9.9 ± 0.5 ^a^	129.7	3.3 ± 0.7 ^a^	97.8	2.6 ± 0.4 ^ab^

The values in the table represent the arithmetic mean (*n* = 8) ± SD (standard deviation). The same letters next to the numbers describe no statistically significant differences between the treatments (Fisher’s LSD test, *p* ˂ 0.05). DM—dry matter, AGB—above-ground biomass.

**Table 3 plants-10-02188-t003:** Crop nitrogen use efficiency.

Treatment	NUE_Crop_	
	%	Rel. %
Control	-	-
UAN	30.5 ± 5.3 ^b^	100.0
UANS1	29.9 ± 7.9 ^b^	97.8
UANS2	50.1 ± 5.6 ^b^	164.2
UANS3	73.4 ± 8.2 ^a^	240.3

The values in the table represent the arithmetic mean (*n* = 8) ± SD (standard deviation). The same letters next to the numbers describe no statistically significant differences between the treatments (Fisher’s LSD test, *p* ˂ 0.05).

**Table 4 plants-10-02188-t004:** Properties of soil used in pot experiment.

Soil Parameter	Value	Ref
pH (CaCl_2_)	6.09	[82]
Soil oxidizable carbon (Cox)	0.80%	[83]
Clay	20%	[84]
Silt	27%	[82]
Sand	53%	[82]
Cation Exchange Capacity	164 mmol/kg	[82]
N total	0.19%	[82]
N-NH_4_^+^ (K_2_SO_4_)	1.48 mg/kg	[82]
N-NO_3_^−^ (K_2_SO_4_)	17.2 mg/kg	[82]
S (water soluble)	8 mg/kg	[82]
P (Mehlich 3)	36.4 mg/kg	[82]
K (Mehlich 3)	400 mg/kg	[82]
Ca (Mehlich 3)	2720 mg/kg	[82]
Mg (Mehlich 3)	214 mg/kg	[82]

**Table 5 plants-10-02188-t005:** Experimental treatments of foliar application in pot experiment.

Treatment	Proportion of Fertilizer in the UANS Mixture		Nutrient Content in the UANS Mixture (Weight %)	
	UAN	S Suspension	N	S
Control	0	0	0	0
UAN	100%	0	30	0
UANS1	66%	33%	20	4
UANS2	50%	50%	15	6
UANS3	33%	66%	10	8

**Table 6 plants-10-02188-t006:** Timetable of the experiment.

Term		Growth Stages of Maize	Operation
1 April 2019		seed	Maize sowing
17 June 2019		4th true leaf	Foliar application of fertilizer mixtures
24 June 2019	t1	5th true leaf	Measurement of chlorophyll content and fluorescence parameters
8 July 2019	t2	6th true leaf	Measurement of chlorophyll content and fluorescence parameters
22 July 2019	t3	7th true leaf	Measurement of AGB harvest, DM weight, content and uptake of nutrient, N:S ratio

**Table 7 plants-10-02188-t007:** Measurement protocol of the chlorophyll fluorescence parameters.

Chlorophyll Fluorescence Parameters	Pulse Type	Light Intensity (μmol/m^2^/s)	Phase	Duration (s)	1st Pulse (s)	Pulse Interval (s)
*Φ_PSII_, F_v_*	Saturation	2400	-	1 pulse		
*R_Fd_*	Flash	900	L	60	0.2	1
			DR	88	1	1
	Saturation	2400	L	60	7	12
			DR	88	11	26
	Actinic	300	L	60	-	-

The measured at wavelength (*λ*) of 454 nm, L—light, DR—dark recovery, *Φ_PSII_*—quantum yield of photosystem II, *R_Fd_*—chlorophyll fluorescence decrease ratio, *F_v_*—variable fluorescence of the dark-adapted leaves.

**Table 8 plants-10-02188-t008:** The photochemical quenching parameters.

Chlorophyll Fluorescence Parameters		Ref.
*F_v_*	*F_m_*−*F*_0_	[87]
*Φ_PSII_*	*F_m_*−*F*_0_/*F_m_*	[88]
*R_Fd_*	*F_d_*/*F_s_*	[89]

*F_0_*—minimal fluorescence from the dark-adapted leaves, *Φ_PSII_*—quantum yield of photosystem II, *R_Fd_*—chlorophyll fluorescence decrease ratio, *F_m_*—maximal fluorescence from the dark-adapted leaves; *F**_d_*—fluorescence decrease from *F**_m_* to *F**_s_*; *F**_s_*—steady-state chlorophyll fluorescence.

**Table 9 plants-10-02188-t009:** The methods for the determination of nutrients in maize AGB.

Nutrient	Method Used	Device Used	Ref.
N	Kjeldal method	Kjeltec 2300 device (Foss Analytical, Hillerød, Denmark)	[92]
S	Optical emission spectrometry	ICP–OES (Spectro, Kleve, Germany)	[93]

## Data Availability

The data presented in this study are available on request from the corresponding author. Due to the nature of this research, participants of this study did not agree for their data to be shared publicly, so supporting data are not available.

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
