# Peer review of "Using Waste Sulfur from Biogas Production in Combination with Nitrogen Fertilization of Maize (Zea mays L.) by Foliar Application"

_plants, 2021, doi:10.3390/plants10102188_

Round 1

Reviewer 1 Report

The reviewed manuscript concerns the effect of foliar application of waste elemental sulfur  from biogas production in combination with conventional liquid fertilizers UAN applied in different ratios.

The results seems to be interesting, but the manuscript needs to be revised before publication.

  • Please provide information about experiment time ...did the authors conducted the experiment only in single series?!
  • Please comment the meteorological conditions during experiment according to Fig.7
  • Section Materials and methods is necessary to put before section Results and discussion.
  • Please provide more information about UAN mixture with waste elemental sulfur. It waste elemental sulfur was well solubility or not? Will authors have recommended to use of this kind of foliar fertilizer to agriculture practice? - I meen to technical solutions to application of this kind of foliar fertilizer?

Minor remarks

Line 213 above a …please put value;

Line 265 – 2 cm diameter sieve;

Line 265 better “soil properties”. In table 4 authors presented not only soil chemical properties but also physical and physicochemical properties.

Recommended

Minor revision  

Author Response

Dear reviewer,

thank you for your very careful review of our paper, and for the comments, corrections, and suggestions. We have tried to take most of them into account when revising the manuscript. We believe that this has significantly improved our manuscript. Specific responses to the comments are provided below; the revised manuscript is attached.

The reviewed manuscript concerns the effect of foliar application of waste elemental sulfur from biogas production in combination with conventional liquid fertilizers UAN applied in different ratios.

The results seems to be interesting, but the manuscript needs to be revised before publication.

Reviewer's comment:

Please provide information about experiment time ...did the authors conducted the experiment only in single series?!

Authors' response:

The experiment was conducted in one series, as is common in vegetation experiments conducted under controlled conditions. The exact timing of the experiment with individual dates is given in Table 6.

Reviewer's comment:

Please comment the meteorological conditions during experiment according to Fig.7

Authors' response:

The manuscript was edited according to the recommendations.

Reviewer's comment:

Section Materials and methods is necessary to put before section Results and discussion.

Authors' response:

In writing the manuscript, we followed the rules outlined in the Instructions for Authors (Microsoft Word template).

Reviewer's comment:

Please provide more information about UAN mixture with waste elemental sulfur. It waste elemental sulfur was well solubility or not? Will authors have recommended to use of this kind of foliar fertilizer to agriculture practice? - I meen to technical solutions to application of this kind of foliar fertilizer?

Authors' response:

The manuscript was edited according to the recommendations. The results of the pot experiment provide a suitable premise for further research in which we plan to conduct field experiments. Their aim will be to verify not only the effect of applying UAN mixtures with waste elemental sulphur under natural field conditions, but also the technical solution of the application (e.g. nozzle type). Based on these results, it will be possible to recommend the application of the mixture to agricultural practice.

Reviewer's comment:

Line 213 above a …please put value;

Authors' response:

The fact that nitrogen to sulfur ratios above a N:S ratio threshold indicates S deficiency is only a general statement. For the evaluation of S nutritional status and prognosis of crop yield, different S species such as organic S, sulfate, total S, and the N:S ratio of various plant parts are determined, usually during the vegetation period and results are interpreted by employing diverse statistical approaches. It is the large variation in experimental conditions and mathematical procedures which make it more or less impossible to compare results from different experiments. Thus the main objective, the reliable deduction of critical values is confronted with major limitations. Important threshold markers for the S supply are: the symptomatological value, which reflects the S concentration below which deficiency symptoms become visible; the critical nutrient value, which stands for the S concentration above which the plant is sufficiently supplied with S for achieving the maximum potential yield or yield reduced by 5 %, 10 %, and 20 %; and the toxicological value, which indicates the S concentration above which toxicity symptoms can be observed.

Reviewer's comment:

Line 265 – 2 cm diameter sieve;

Authors' response:

The manuscript was edited according to the recommendations.

Reviewer's comment:

Line 265 better “soil properties”. In table 4 authors presented not only soil chemical properties but also physical and physicochemical properties.

Authors' response:

The manuscript was edited according to the recommendations.

Reviewer 2 Report

Dear authors,

Firstly, I would like to thank You for Your interesting work about Using waste sulfur from biogas production in combination 2 with nitrogen fertilization in maize production.

There are several points that I would like to comment:

What about the organic matter content in the soil?

Introduction

Line 52: … so therefore is a sortage of sulfur in the soils – You need to be specific. There are not many soils with deficit of S. Even opposite, the acid soils have enough S. How you can conclude it? Do you have some data of soil analysis in Your area?

In the introduction You do not mentioned nothing about the sulfur in the soil – this should be added in one paragraf.

Meterial and methods

How often did the post watered (line 277, 278)

You chose only three parameters, what about other parameters which PAR-FluorPen FP 110-LM/S  can measure chlorophyll fluorescence parameters (Ft, QY, NPQ, OJIP..)

There is only 1 hybrid included in the study, it would be more interesting if there were

Conclusion

I would like You ti include in Your conclusion – since the addition of sulfur may cause a reduced pH and soil acidification, many of soils in your region I suggest have enough S. What do You think for the future generations – does it is a dobut that adding a sulfur may increase the areas with acid soils?

Author Response

Dear reviewer,

thank you for your very careful review of our paper, and for the comments, corrections, and suggestions. We have tried to take most of them into account when revising the manuscript. We believe that this has significantly improved our manuscript. Specific responses to the comments are provided below; the revised manuscript is attached.

Firstly, I would like to thank You for Your interesting work about Using waste sulfur from biogas production in combination 2 with nitrogen fertilization in maize production.

Reviewer's comment:

What about the organic matter content in the soil?

Authors' response:

The organic matter content of the soil was characterised only as oxidizable carbon content before the establishment of the pot experiment (see Table 4).

Reviewer's comment:

Introduction

Line 52: … so therefore is a sortage of sulfur in the soils – You need to be specific. There are not many soils with deficit of S. Even opposite, the acid soils have enough S. How you can conclude it? Do you have some data of soil analysis in Your area?

Authors' response:

The manuscript was edited according to the recommendations.

Reviewer's comment:

In the introduction You do not mentioned nothing about the sulfur in the soil – this should be added in one paragraph.

Authors' response:

The manuscript was edited according to the recommendations.

Reviewer's comment:

Material and methods

How often did the post watered (line 277, 278)

Authors' response:

Watering was done so that the plants had optimal moisture conditions throughout the growing season. The plants were watered taking into account the external conditions (temperature, humidity) and the growth stages of the plants (water requirement). The amount of water and the intensity of watering therefore varied over time. Therefore, we chose to determine the watering requirement by determining the weight of reference plant pots and plants were watered to 70% maximum water holding capacity throughout the growing season (as stated in the manuscript).

Reviewer's comment:

You chose only three parameters, what about other parameters which PAR-FluorPen FP 110-LM/S  can measure chlorophyll fluorescence parameters (Ft, QY, NPQ, OJIP..)

Authors' response:

Selected fluorescence parameters of chlorophyll that are commonly used to assess the effect of foliar fertilizer application on photosynthesis were used in the manuscript. Other parameters (Ft, NPQ, OJIP, …) were also determined with the instrument, but mostly correlated with the parameters we presented. We believe that the photosynthesis parameters (The variable fluorescence of the dark-adapted leaves - Fv, Quantum yield of photosystem II - ΦPSII and Chlorophyll fluorescence decrease ratio - RFd) used are sufficient to assess the effect of application on photochemical efficiency of photosystem II.

Reviewer's comment:

There is only 1 hybrid included in the study, it would be more interesting if there were

Authors' response:

The experiment was set up as a pilot to test the effect of foliar application of fertilizer mixtures (UAN and waste sulphur), so far only on one maize hybrid. Moreover, the effect of the application was only observed in a part of the vegetation, therefore we did not expect the effect of "variety (hybrid)" to be demonstrated in the experiment with the design used. Also, given the range of measurements and the number of analyses performed, only one maize variety was used. We plan to test the effect of fertilizer mixture application under field conditions where we will include the effect of variety.

Reviewer's comment:

I would like You ti include in Your conclusion – since the addition of sulfur may cause a reduced pH and soil acidification, many of soils in your region I suggest have enough S. What do You think for the future generations – does it is a dobut that adding a sulfur may increase the areas with acid soils?

Authors' response:

The effect of applying elemental sulphur to the soil increases soil acidification, especially depending on soil type (light, heavy soil) and sulphur dose. These conclusions are documented by many studies (e.g. Tabak, M.; Lisowska, A.; Filipek-Mazur, B.; Antonkiewicz, J. The Effect of Amending Soil with Waste Elemental Sulfur on the Availability of Selected Macroelements and Heavy Metals. Processes 2020, 8, 1245.). However, we believe that foliar application of sulphur (in addition to nitrogen fertilizers) eliminates these negative impacts. Since determining the effect of foliar application of S on soil acidity was not of interest to us, we did not discuss this issue in the manuscript.

Reviewer 3 Report

This paper describes an experiment where S recovered from biogas production was used in foliar fertiliser application. It is investigated how it affects growth, N and S uptake and photosynthesis. The idea of using this residue is good, and the paper should be published on that merit. It would have been extremely interesting also to see some analysis of the economy in this – Is it likely that this could be a technology that would be used?

However, it appears that few effects of S application rate could be found, this should at least be not be hidden. It is possible that using a soil lower in nutrients would have given a stronger effect. However, some effects were found, and certainly no harmful effects were found.

The authors should aim at writing a clearer manuscript where the main effects are discussed first, and then the minor. And it should be made clear when the results from the present study is in contract to or in agreement with previous studies mentioned.

Specifically: I assume that this journal prescribes that results and discussion should come before materials and methods? This is otherwise somewhat unusual.

Abstract: The description of results is confusing. Do not try to hide that few effects were found. Introduction: I would like to have a bit more background about foliar application – under what circumstances is it used, and what are advantages and disadvantages? A bit more theory on the interaction between S and N fertilisation would also be recommended.

Results and discussion: How was it decided what should be shown as tables and what should be shown as figures? Usually I would assume that the most important and/or the ones that show differences would be shown as figures.

I would also recommend to discuss biomass yield and N and S uptake first, even though this did not show differences. Explain more what you think the differences that were observed mean, and try to shorten and simplify the discussion. It is not necessary to mention all results in the text if they are in figures and tables, just mention those you want to discuss. When other studies are referred to, make it clear if it supports or is in contract to the present study.

I believe there should be a conclusion section, if this is not allowed by journal, it could still be added as a paragraph.

Materials and methods: Figure 8 is not needed, it is sufficient to say that pots were placed in a random block design. I do not understand much of the photosynthesis measurements, but then it is also not my field.

Author Response

Dear reviewer,

thank you for your very careful review of our paper, and for the comments, corrections, and suggestions. We have tried to take most of them into account when revising the manuscript. We believe that this has significantly improved our manuscript. Specific responses to the comments are provided below; the revised manuscript is attached.

Reviewer's comment:

This paper describes an experiment where S recovered from biogas production was used in foliar fertiliser application. It is investigated how it affects growth, N and S uptake and photosynthesis. The idea of using this residue is good, and the paper should be published on that merit. It would have been extremely interesting also to see some analysis of the economy in this – Is it likely that this could be a technology that would be used?

Authors' response:

Waste sulphur used in UAN fertiliser mixtures does not significantly increase the cost of fertilisation. The price of waste sulphur is lower than the price of UAN fertiliser, so the cost of fertiliser (mixtures) with an increased proportion of waste sulphur is lower. Foliar application of UAN fertiliser is a common addition to the nutrition of many crops and the combination of this fertiliser with waste sulphur is both economical and environmentally friendly.

Reviewer's comment:

However, it appears that few effects of S application rate could be found, this should at least be not be hidden. It is possible that using a soil lower in nutrients would have given a stronger effect. However, some effects were found, and certainly no harmful effects were found.

Authors' response:

We are not trying to hide anything. We have tried to comment properly on all the results presented in the study. The main objective of our work was to verify the effect of foliar application of mixtures of UAN fertilizer and waste sulphur under controlled conditions. We believe that the results of pilot verification will serve as a good basis for field experimentation.

Reviewer's comment:

The authors should aim at writing a clearer manuscript where the main effects are discussed first, and then the minor. And it should be made clear when the results from the present study is in contract to or in agreement with previous studies mentioned.

Authors' response:

We believe that incorporating the Discussion into the Results part of manuscript is a more appropriate way of interpreting the results rather than dividing the text into two parts Results and Discussion. The results were described chronologically as they were achieved during the experiment (in a logical time sequence).

For the results of other studies, we tried to adapt the text as recommended.

Reviewer's comment:

Specifically: I assume that this journal prescribes that results and discussion should come before materials and methods? This is otherwise somewhat unusual.

Authors' response:

In writing the manuscript, we followed the rules outlined in the Instructions for Authors (Microsoft Word template).

Reviewer's comment:

Abstract: The description of results is confusing. Do not try to hide that few effects were found.

Authors' response:

As already stated, it is not our intention to hide any results. In writing the manuscript, we follow the author's instructions, which require us to write an abstract of no more than 200 words. Our aim was to present the most important results in the abstract and at the same time to present it in the structure given by the instructions to the authors.

Reviewer's comment:

Introduction: I would like to have a bit more background about foliar application – under what circumstances is it used, and what are advantages and disadvantages? A bit more theory on the interaction between S and N fertilisation would also be recommended.

Authors' response:

The manuscript was edited according to the recommendations.

Reviewer's comment:

Results and discussion: How was it decided what should be shown as tables and what should be shown as figures? Usually I would assume that the most important and/or the ones that show differences would be shown as figures.

Authors' response:

The results in the tables and figures always follow the same format (mean value per treatment, standard deviation and a letter indicating significance). Therefore, we believe that it does not matter whether the result is presented in the form of a table or a figure. If the reviewer insists on modification, there is no problem in modifying the results into a form that suits him.

Reviewer's comment:

I would also recommend to discuss biomass yield and N and S uptake first, even though this did not show differences. Explain more what you think the differences that were observed mean, and try to shorten and simplify the discussion. It is not necessary to mention all results in the text if they are in figures and tables, just mention those you want to discuss. When other studies are referred to, make it clear if it supports or is in contract to the present study.

Authors' response:

As noted, in the section "Results and Discussion", we have attempted to both describe and discuss the results chronologically as they were laid out over time. We believe that the sources used in the discussion are directly related to the results obtained and many of them help to clarify and explain them. Based on the recommendations, we sought to add to the studies used in the discussion whether they support or refute the results of our research.

I believe there should be a conclusion section, if this is not allowed by journal, it could still be added as a paragraph.

Authors' response:

At the end of the section “Results and Discussion” was a paragraph evaluating the pot experiment. This was edited and placed at the end of the manuscript under the section “4. Conclusions”. 

Reviewer's comment:

Materials and methods: Figure 8 is not needed, it is sufficient to say that pots were placed in a random block design. I do not understand much of the photosynthesis measurements, but then it is also not my field.

Authors' response:

The figure 8 has been removed.
